# Antibacterial Effect of Carbon Nanomaterials: Nanotubes, Carbon Nanofibers, Nanodiamonds, and Onion-like Carbon

**DOI:** 10.3390/ma16030957

**Published:** 2023-01-19

**Authors:** Ekaterina Moskvitina, Vladimir Kuznetsov, Sergey Moseenkov, Aleksandra Serkova, Alexey Zavorin

**Affiliations:** 1Siberian Federal Research and Clinical Center of FMBA of Russia, 636000 Tomsk, Russia; 2Boreskov Institute of Catalysis SB RAS, 630090 Novosibirsk, Russia

**Keywords:** carbon nanotubes, nanodiamond, catalytic filamentous carbon, onion-like carbon, antibacterial effect, *Escherichia coli*, *Staphylococcus aureus*

## Abstract

The increasing resistance of bacteria and fungi to antibiotics is one of the health threats facing humanity. Of great importance is the development of new antibacterial agents or alternative approaches to reduce bacterial resistance to available antibacterial drugs. Due to the complexity of their properties, carbon nanomaterials (CNMs) may be of interest for a number of biomedical applications. One of the problems in studying the action of CNMs on microorganisms is the lack of universally standardized methods and criteria for assessing antibacterial and antifungal activity. In this work, using a unified methodology, a comparative study of the antimicrobial properties of the CNM systemic kit against common opportunistic microorganisms, namely *Escherichia coli* and *Staphylococcus aureus*, was carried out. Multiwalled carbon nanotubes (MWNTs), catalytic filamentous carbon with different orientations of graphene blocks (coaxial–conical and stacked, CFC), ionic carbon (OLC), and ultrafine explosive nanodiamonds (NDs) were used as a system set of CNMs. The highest antimicrobial activity was shown by NDs, both types of CFCs, and carboxylated hydrophilic MWCNTs. The SEM results point out the difference between the mechanisms of action of UDD and CFC nanotubes.

## 1. Introduction

The discovery of antibiotics caused a dramatic decline in morbidity and mortality and revolutionized medicine in the early twentieth century. However, the effectiveness and easy access to antibiotics have also led to their overuse and the spread of antibiotic tolerance and resistance [1,2]. Currently, the bacterial world has demonstrated the ability to quickly adapt to antibiotics using a wide range of mechanisms [3,4,5]. Already, physicians are confronted with infectious agents that are resistant to one, several, or all classes of antibiotics [6]. The World Health Organization has named the rise in bacterial resistance to antibiotics as one of the ten global health threats facing humanity. Despite the development of new antibiotics, the resistance of bacteria and fungi, particularly to new drugs, is steadily growing [7]. This naturally affects the effectiveness of antimicrobial therapy, the duration of treatment, and, in some cases, the outcome of infectious diseases. Therefore, the development of new antibacterial agents or the use of innovative strategies aimed at improving pharmacokinetics and reducing bacterial resistance to available antibacterial drugs is of great importance.

The discovery of carbon nanomaterials (CNMs)—such as carbon nanotubes (CNTs), graphene, fullerenes, and nanodiamonds, which are characterized by extremely small sizes and, therefore, a developed surface area and the ability to vary the composition of surface functional groups—opens up new opportunities for the development of materials for use in medicine [8,9,10]. In particular, one of the promising areas of research on carbon nanomaterials is the biomedical application of nanoparticles since they can be considered as primary building blocks for multifunctional nanotools intended for future therapeutic and diagnostic purposes [11,12]. It was found that a number of carbon nanostructures have bactericidal and fungicidal properties in relation to pathogenic microorganisms [13,14,15], which can open up new alternative ways in the fight against pathogens of bacterial and fungal infections. 

In various studies, the antimicrobial activity of CNMs, such as single-layer and multilayer purified and functionalized carbon nanotubes, various types of nanodiamonds, etc., has been repeatedly investigated [16,17,18], but this area is still poorly understood. In addition, one of the problems is the lack of universally standardized methods and criteria for assessing antimicrobial activity. For example, the influence of factors such as the size and tendency to aggregate of CNMs, contact time, the action of buffer solutions, and the ability of the bacterial cell to produce substances that block the bactericidal action of CNMs (for example, exopolysaccharides that form a bacterial film) can give different final results, even when studying the same CNMs [19,20,21].

However, it is known that the main mechanisms of the bactericidal action of CNMs are due to a combination of their physical and chemical properties. Physically, CNMs can cause significant structural damage to the cell wall and membrane of a microorganism. In addition, they have the ability to biologically isolate the cell from the microenvironment, which ultimately leads to the production of toxic substances such as reactive oxygen species and exposes the cell to oxidative stress, leading to its biological death [22,23]. It is noteworthy that CNMs have different mechanisms of action than antibiotics and exhibit different activity, including against those bacteria that have already acquired antibacterial resistance [24,25]. Because of this, CNMs may be of interest in the development of new antibacterial applications.

In this study, within a single methodology, we sought to determine the presence of antimicrobial properties in a wide systemic set of CNMs and compare the effects caused by different types of CNMs on typical common opportunistic pathogens, namely *Escherichia coli* (hereinafter *E. coli*) and *Staphylococcus aureus* (hereinafter *S. aureus*).

## 2. Materials and Methods

### 2.1. Nanomaterials 

A system set of nanostructured carbon materials has been used in this work (Table 1): **(1)****Nanodiamonds** (ND, sp3-hybridized carbon material, primary particle mean size 5 nm, Appendix A) were produced by FGUP “Altay” (Biysk, Russia) using detonation of a trotyl/cyclotrimethylene–trinitroamine mixture (TNT/RDX = 50:50) according to the procedure described elsewhere [26] and purified with a mixture of nitric and sulfuric acids [27]. **(2)****Onion-like carbons** (OLCs, Appendix A) were prepared via the ND thermal treatment at 1527 °C under vacuum. OLCs consist of defective closed graphite shells enclosed in each other [28,29]. **(3)****Catalytic filamentous carbon** (CFC-I, Appendix A) was produced through methane decomposition over a 90Ni–Al_2_O_3_ catalyst. In CFC-1, the graphite-like planes are oriented coaxially, with the angle between the planes and the filament axis varying from 45 to 75° [30,31]. **(4)****CFC-2** (Appendix A) was produced via methane decomposition over a 75Ni–15Cu–Al_2_O_3_ catalyst. CFC-2 consists of graphite-like planes stacked perpendicular to the filament axis [31]. **(5)****A set of multiwalled carbon nanotubes** (MWCNTs, Appendix A) with variable external diameters and lengths (Table 1) was produced using the CCVD technique in the reaction mixture of C_2_H_4_ and Ar (50:50) at 650–670 °C with Fe–Co catalysts [32,33,34,35,36]. Primary MWCNTs exist in the form of agglomerates consisting of randomly entangled nanotubes [35]. To remove catalyst residues, the MWCNT samples were boiled in diluted HCl (1:1) for 2 h, which was followed by washing with distilled water to neutral reaction and air-drying at 80 °C for 2 days. To obtain short MWCNTs, milling in an APF-type plantar mill was used. Further in the text, the following constructions are used to designate short nanotubes: in the notation MWNT-A (L), A (values 1,2,3) refer to nanotubes with an average diameter of 7.8, 10, and 18 nm, respectively; the numbers in brackets (L) correspond to the average length of nanotubes in µm. Thus, MWCNT-2 (0.6) corresponds to nanotubes with an average diameter of 10 nm and a length near 0.6 μm.

**Carboxyl-functionalized MWCNTs (MWCNT-ox)** were prepared by boiling in concentrated nitric acid for 2 h. X-ray photoelectron spectroscopy (ES-300, Kratos Analytical, Shimadzu Corporation of Kyoto, Japan) showed that the surface of oxidized MWCNTs contains the following oxygen-containing groups: carboxyl (34.3%), hydroxyl (27.9%), and carbonyl (37.8%) groups [37]. 

**Preparation of suspensions of carboxylated MWCNT-ox**. Aqueous suspensions of MWCNT-1ox (0.6) and MWCNT-2ox (0.5), which were preliminarily ground by mechanical treatment in an APF-3 planetary mill (Novic, Novosibirsk, Russia), were prepared via ultrasonic (US) treatment of 0.5 g MWCNTs (a Volna ultrasonic disperser, 2 h, 400 W, 22 kHz) in 1.5 L of distilled water followed by settling for 3 days. The concentration of MWCNTs in the suspension was estimated after settling by drying the suspensions. The resulting stable suspensions contained 0.267 mg/mL of MWCNT-1ox (0.6) and 0.317 mg/mL of MWCNT-2ox (0.5), respectively. 

**Preparation of CNM suspensions**. Before use, weighed portions of CNM powders (10 mg) were dispersed in 100 mL of distilled water in an ultrasonic bath (Ultrasonic VGT 1000, 35 W, China) for 30 min to obtain suspensions with a concentration of 100 µg/mL. It should be noted that the stability of suspensions during their settling was different (Figure 1). The most stable suspensions (more than 1 day) were formed by CFCs, NDs, and MWCNT-ox due to the presence of oxygen-containing groups on their surfaces. However, for all the CNMs, when testing their antibacterial activity in vitro, constant stirring of the suspensions in a shaker was used.

**Titrimetric analysis of oxygen-containing groups** on the surface of MWNTs was performed using the reverse acid-base titration technique described by Boehm [38].

**Temperature-programmed desorption spectra** (mass-spectrometric analysis of the evolution of H_2_O, CO, and CO_2_ during heating of CFC in vacuum) of oxygenated groups of CNMs were monitored using a QMS300 quadrupole mass spectrometer (Stanford Research Systems, Stanford, CA, USA) in the range of m/z ¼ 1-65, with a resolution of 0.1 (H_2_O (m/z ¼ 18), CO (m/z ¼ 28), CO_2_ (m/z ¼ 44)) [39].

### 2.2. Microorganisms

Standard reference strains of the american type culture collection E. coli ATCC 25922 and S. aureus ATCC 25923 (representatives of gram-negative and gram-positive bacteria) were used in the study. Prior to the experiments, microbe cultures were grown in LB broth (Miller) at 37 °C for 20∼24 h and harvested during mid-exponential growth phase. Cells were washed by centrifugation and resuspended in sterile deionized water three times. Cell suspensions were then diluted to the desired concentration.

### 2.3. Media and Cultivation Conditions 

Isolation of bacterial strains was carried out via classical culture methods on conventional nutrient media Columbia agar, Endo medium, and staphylococcus agar. As it is known, CNMs, when interacting with various substances including solutions, can change the size of aggregates with a corresponding change in their properties. For example, the presence of NaCl in a solution leads to a stronger aggregation of nanodiamonds [40]. To exclude the influence of impurities, the antimicrobial properties of CNMs were evaluated in distilled water.

### 2.4. A Study of the Antimicrobial Efficacy of Nanoparticles In Vitro 

To assess the effect of CNMs on the microorganism, the studied bacteria were suspended in deionized water, and brought to a turbidity level corresponding to 0.5 McFarland (1.5 × 10^8^ CFU mL, colony-forming units). Then, 0.20 µL of the microbial suspension was added to 280 µL of CNM, preliminarily diluted to a concentration of 100 µg/mL (except when using a suspension of carboxylated MWCNTs). The choice of these indicators was based on the fact that most of the studies reported in the literature used CNM concentrations for the analysis of antimicrobial activity in the range of 50 to 500 μg/mL. The resulting mixtures were incubated in the dark using a Bact/Alert (RF) analyzer maintained at 37 °C and stirring at 60 oscillations per minute. After a day of incubation under these conditions, 0.20 µL of suspension was inoculated on nutrient media in Petri dishes, and after 24–48 h the number of colonies was counted. Each assay was repeated three times for each CNM and microorganism tested. The graphs represent the mean of the three results obtained ± standard deviation. The experiments were accompanied by a control with microorganisms that did not contain CNM.

After identifying the CNM with the greatest antimicrobial effects, the dependence of the antibacterial properties of the CNM on the exposure time was determined. To do this, we used the same technique described above but successively reduced the exposure time and then the concentration of the investigated CNMs from 100 μg/mL to 30 and 10 μg/mL. For kinetic studies, *E. coli* was used as a model.

### 2.5. Microscopic Methods

Light luminescence microscopy was performed using Nikon Eclipse E200 (Nikon, Tokyo, Japan) and Carl Zeiss Axiostar microscopes (Carl Zeiss, Jena, Germany). Luminescence microscopy slides were stained with fluorochromes propidium iodide (PI), acridine orange, and SYTO™ 9. Scanning electron microscopes (JSM6460-LV JEOL and SU8230 Hitachi, Japan) were used to study the interaction products of bacteria with CNMs. Suspensions of the CNM mixtures with microorganisms were applied to aluminum foil and, after drying, were placed in the working chamber of the microscope. To obtain SEM images of bacteria adsorbed on CNMs, we did not use any contrasting or stabilizing agents. The structure of MWCNTs was also characterized with transmission electron microscopy (TEM, JEM 2010, JEOL, Tokyo, Japan) and Themis-Z 3.1 (Thermo Fisher Scientific, Eindhoven, Netherlands). For TEM characterization of the specimen structure, a sample was deposited on a copper grid with a carbon film. Nanotube average diameters were estimated from several TEM images taken at a low ×100,000 magnification.

**The specific surface area** (S_BET_) of the samples was monitored using N_s_ adsorption isotherms (77 K) obtained with a surface area and porosimetry analyzer ASAP-2400 (Micromeritics, Norcross, GA, USA).

## 3. Results and Discussion

### 3.1. Determination of the Antimicrobial Efficacy of CNMs

Figure 2 and Figure 3 display typical images of the Petri dishes used to determine CFUs after treatment of bacteria with CNM suspensions (0.5 McFarland, 1.5 × 10^8^ CFU/mL) according to Section 2.4. The statistically processed results of the survival of *E. coli* and *S. aureus* after exposure to CNM are shown in Figure 4.

#### 3.1.1. CFCs 

Suspensions of CFCs with different orientations of graphene blocks (coaxial–conical and stacked) completely suppressed the viability of *E. coli* and *S. aureus*. It can be seen from the SEM microscopy data that bacterial cells attach to CFC agglomerates with further damage and cell death (Figure 5A,B, for *E. coli*). Apparently, such aggregation of bacteria with nanofibers leads to the blocking of basic cellular functions with the destruction of cell membranes. This is confirmed through the data from the fluorescence microscopy with staining of cells with SYTO 9 dye (Figure 5C,D). It should be noted that the sizes of the fragments of destroyed bacteria were much smaller in the interaction of CFCs with S. aureus than in the case of *E. coli* (not presented here).

In the literature, we did not find data on the antimicrobial effects of CFCs. However, at a concentration of 100 μg/mL, CFCs were already 100% effective in suppressing the viability of *E. coli* within an hour, as evidenced via fluorescence microscopy and the absence of growth on agar media. With a decrease in the concentration of CFCs, the antimicrobial effect expectedly decreased (Figure 6). It should be noted that CFC-2 with stacked orientations of graphene blocks turned out to be the most effective, which is probably due to a higher surface concentration of edges of graphene fragments.

#### 3.1.2. MWCNTs

Of the multiwalled nanotubes from the studied set, carboxylated MWNTs turned out to be the most effective, almost completely suppressing the viability of *E. coli* and *S. aureus*. As can be seen from luminescence, SEM, and light microscopy, nanotubes are capable of both adsorbing on aggregates and partially or completely enveloping the microorganism cells, damaging the cell wall. When completely enveloping, nanotubes prevent the metabolism and, as a result, biological isolation and death of bacteria occur [41,42,43,44]. With partial interaction, MWCNTs damage the cell wall and membrane-like barbed wire [45,46,47]. Of the non-functionalized tubes, tubes with a smaller diameter and length (MWCNT-1 (0.6)) turned out to be the most active ones. At the same time, stable suspensions of thinner nanotubes (MWCNT-1(2)ox) exhibited the highest activity among carboxylated MWCNTs. Obviously, the smaller the particle size of the tubes suspended in the solution and the more of them there are per unit weight, the stronger the physical effects of damage to bacterial cells will be. It is this mechanism of the antimicrobial effect of all MWCNTs that we consider to be the main one, which is also confirmed via SEM microscopy (Figure 7).

When studying the time of exposure to carboxylated MWCNTs, inoculation was performed with a pure suspension of *E. coli* (control) and a suspension of *E. coli* and MWCNTs after 1, 4, 8, 16, and 24 h. The results showed that the suppression of the ability to form colonies reached a maximum after 16 h (Figure 8).

#### 3.1.3. Nanodiamonds

Nanodiamonds have been used as an antibacterial material in other studies [48,49,50], which show the importance of nanoparticle size, oxidation state, and charge of surface groups. It is interesting that NDs with a purity class of 01, a negative zeta potential, and the presence of oxygen-containing groups on their surface were especially effective against both the Gram-negative and Gram-positive bacteria, while highly purified NDs obtained by the detonation method were no longer able to exhibit antibacterial properties [51]. In our work, NDs completely suppressed the growth of *E. coli* and *S. aureus*. At the same time, SEM microscopy shows that bacterial cells adhere to agglomerates, resulting in further damage and cell death (Figure 9). It is important to note that keeping the interaction products of *E. coli* and *S. aureus* bacteria in an aqueous medium for 3 weeks led to the almost complete disappearance of bacterial cell residues on the ND surface. At the same time, on the dried samples, the remains of bacterial cells were stable for at least 2 months. 

When studying the exposure time, it turned out that NDs, like CFCs, completely suppress the ability to form colonies within one hour. As the concentration of NDs decreased, the antimicrobial efficacy also decreased (Figure 10).

#### 3.1.4. OLC

Despite the huge interest in various forms of carbon nanostructures, carbon nano-onions have received much less attention so far. Toxicological data for OLCs are very scarce [52]. In our study, carbon OLCs showed no antimicrobial properties against either microorganism, which is consistent with another recent study of OLC toxicity in animal, human, and *E. coli* cells, in which OLCs were inert to *E. coli* and did not induce cell secretion involved in inflammation and immune responses (interleukins 2 (IL-2) and tumor necrosis factors) [53]. This behavior of OLCs can be explained by the absence of defects and functional groups on their surface due to the high-temperature treatment used during their preparation, which leads to the formation of closed fullerene-like shells. It should be noted that, due to their hydrophobicity, OLCs do not give sufficiently stable suspensions, which also significantly reduces the possibility of contact with the surface of bacterial cells.

### 3.2. Influence of the Content of Oxygen-Containing Groups on the Antibacterial Effects of CNMs

It is well known that organic acids inhibit the vital activity of microbes. In particular, the MIC (minimum inhibitory concentration) of citric acid for *E. coli* and *S. aureus* was 0.06 g/mL [54]. Because of this, one of our assumptions about the mechanism of bactericidal properties was the presence of active oxygen-containing groups on the surfaces of CNMs (namely, NDs, CFCs, and carboxylated MWCNTs). After heating the most active CNMs at 300, 500, and 800 °C in a hydrogen flow, the oxygen groups were sequentially removed, which was confirmed via the analysis of desorbed gases (Figure 11).

We repeated experiments on the effect of decarboxylated CNMs (heated in a hydrogen flow) on *E. coli* and obtained results similar to those observed in the presence of CNMs with oxygen groups on the surface (Table 2). These experiments exclude the mechanism of antimicrobial action due to the presence of oxygen-containing groups on the CNM surface.

### 3.3. Influence of the Liquid Medium on the Antimicrobial Effects of CNM

We also studied the dependence of the antibacterial action on the liquid medium in which the CNM and bacteria were placed. To do this, the standard experiments were repeated, but the nutrients 1% peptone and 0.5% glucose (a complete medium for bacteria’s growth) were added as the basis for the suspension in deionized water. Peptone is a complex mixture of proteins and amino acids obtained from the enzymatic hydrolysis of animal tissues. After adding nutrients to the *E. coli* and CNM medium, the suspensions were incubated at 370 °C on a shaker, followed by seeding and counting CFU after 30 min, 1 h, and 24 h. The experiments were accompanied by a control with bacteria without the addition of CNM. It was found that, as a result of adding the nutrients necessary for bacterial growth to the suspension, the long-term antimicrobial effects disappeared for all the CNMs (Figure 12).

This effect can be explained, on the one hand, by the competitive adsorption of proteins, amino acids, sugar, and bacterial cells, and, on the other hand, by a change in the chemistry of the CNM surface, which leads to a decrease in cell membrane damage in the process. In addition, the reproduction rate of *E. coli* is one fission every 20 min. Within an hour, the number of microorganisms under favorable conditions can triple. It is known that the ratio of concentrations of microorganisms and substances interacting with them is a key factor in the manifestation of bactericidal properties. For highly dispersed materials, their antimicrobial potential is determined through the concentration of surface centers interacting with microorganisms. Upon direct physical contact of CNM agglomerates with microorganisms during competitive adhesion between the cell wall and complex organic molecules contained in the nutrient medium, a certain part of the microorganisms does not interact with the CNM surface or interacts without causing damage sufficient for death, which results in the survival and further reproduction of cells (Figure 13).

## 4. Conclusions

In our study, we compared the antibacterial properties of a systemic set of carbon nanomaterials with different structures using a single method, which seems to be the simplest and most reliable one. We have shown that the antimicrobial effect of CNMs strongly depends on their structure, particle size, degree of aggregation, concentration, and surface functionalization. Thus, onion-like carbons did not show antimicrobial properties, possibly due to the low concentration of surface defects and hydrophobicity, which resulted in their low ability to adsorb microorganisms. 

At the same time, nanodiamonds, catalytic filamentous carbon, and, to a lesser extent, carboxylated MWCNTs have an antimicrobial effect. Under the influence of NDs and CFCs, bacterial cells stick together to form agglomerates, resulting in the destabilization of the cell wall and membrane. Moreover, with prolonged contact, almost complete destruction of the interaction products of bacterial cells with these CNMs is observed. One of the main mechanisms of action of MWCNTs is physical damage to cells; the better and more stable they are dispersed in the MWCNT solution, the more pronounced their bactericidal properties. The authors of the review [13], after analyzing a large number of articles, concluded that the toxic dose of CNTs IC50 is about 4–5 mg mL^−1^ h^−1^. Based on our data, we also suggest that the toxicity range of MWCNT-ox is comparable to this value. 

We also conducted studies both in water and in nutrient broth, which showed the dependence of the antimicrobial action on the composition of the solutions, which is explained by the possibility of blocking the centers that firmly bind microorganisms and cause the destruction of their walls.

These observations indicate the need to develop antibacterial systems using several components. In particular, combinations of CNMs with coated silver particles or antibiotics may be of interest. At the same time, the identified CNMs with a high bactericidal effect can also be used as components of air-cleaning devices, membranes, antibacterial fabric materials, and polymer coatings, where the influence of liquid media is significantly limited.

It should be mentioned that in this study we did not consider the issues of nanotoxicity of the used NCMs [55,56,57,58], since the aspects of their further application, which we are developing, do not involve the introduction of their powders into the human body or into the environment. Thus, we assumed the development of devices that provide air disinfection (membranes, filter materials) and antimicrobial coatings. It is supposed to provide a sufficiently strong bond between the NCMs and the structural materials of the base. In these cases, even with the destruction of such materials, the toxicity of the enlarged products of their destruction (compared to primary CNMs) will be close to the low toxicity of widespread carbon materials (soot, coals) with which mankind has been in contact throughout its history.

## Figures and Tables

**Figure 1 materials-16-00957-f001:**
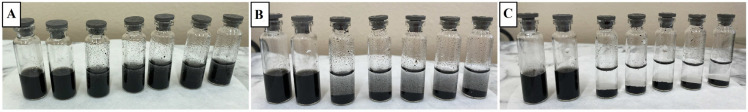
Stability of aqueous suspensions of MWCNTs. (**A**) Immediately after dispersing in an ultrasonic bath, (**B**) after 1 h exposure, (**C**) after a week. From left to right: MWCNT-1-ox (0.6), MWCNT-2-ox (0.5), MWCNT-1, MWCNT-1 (0.6), MWCNT-2, MWCNT-2 (0.5), and MWCNT-3.

**Figure 2 materials-16-00957-f002:**
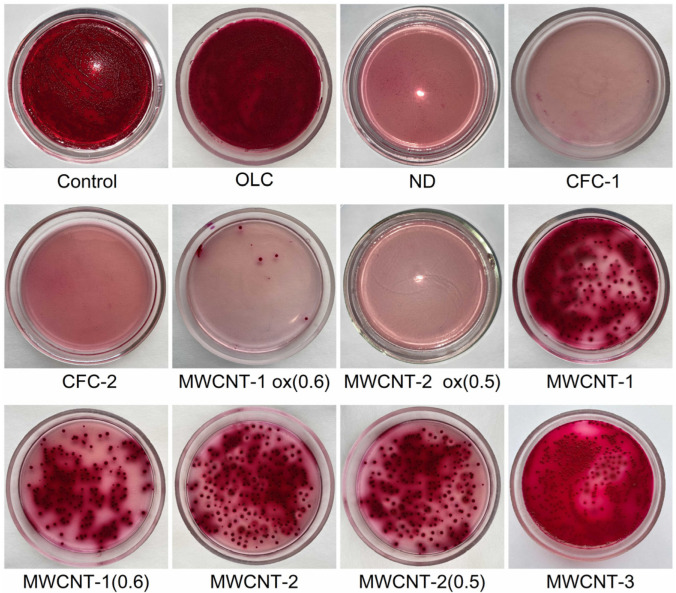
Typical examples of plates (with *Endo agar*) used to determine the number of microbial cells (CFU) after exposure to CNMs on *E. coli*.

**Figure 3 materials-16-00957-f003:**
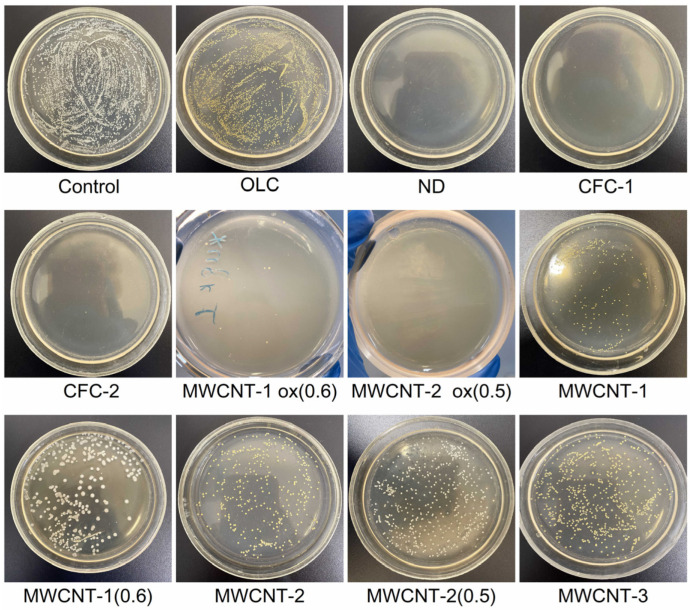
Typical examples of plates (with *Staphylococcus agar*) used to determine the number of microbial cells (CFU) after exposure to CNMs on *S. aureus*.

**Figure 4 materials-16-00957-f004:**
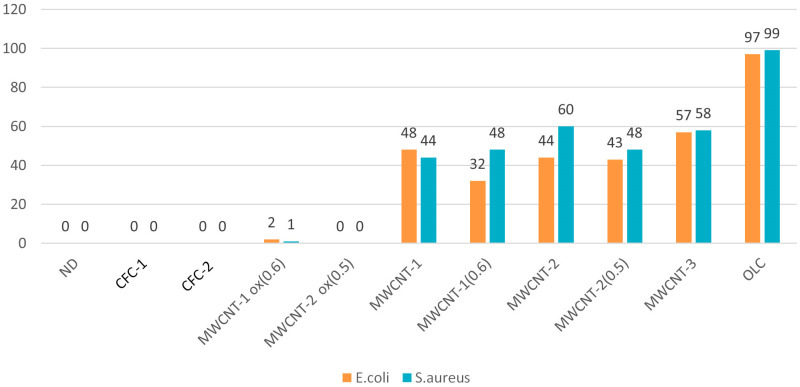
Antimicrobial activity of CNM suspensions (230 µL of 100 µg/mL suspension) on *E. coli* and S. aureus (0.20 µL of 1.5 × 10^8^ CFU/mL suspension).

**Figure 5 materials-16-00957-f005:**
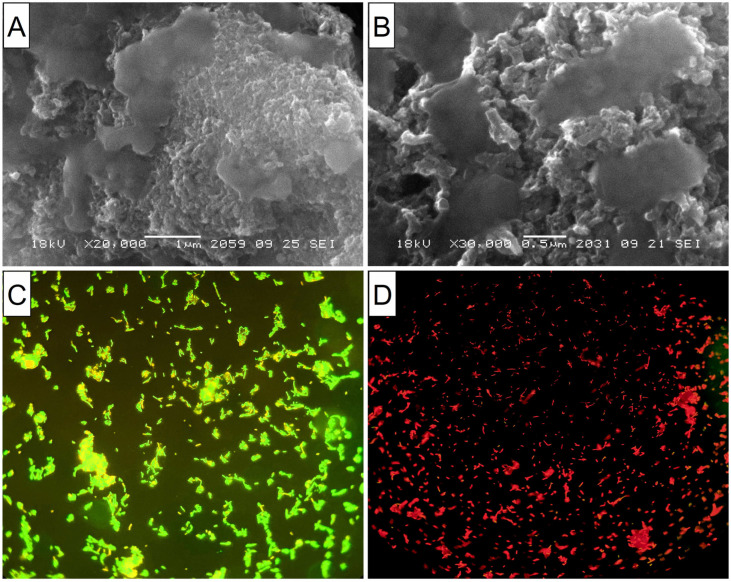
(**A**,**B**) SEM images of CFC-1 aggregates with fragments of destroyed *E. coli* cells. (**C**,**D**) Fluorescence microscopy images. SYTO 9 stain and the propidium iodide stain live bacteria with intact cell membranes fluoresce green, while dead bacteria with compromised membranes fluoresce red. (**C**) Live *E. coli* without exposure to CNFs. (**D**) Dead cells after 24 h of exposure to CFC-1 (a ×400 magnification).

**Figure 6 materials-16-00957-f006:**
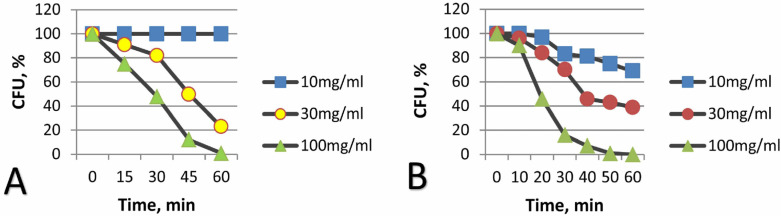
Influence on the kinetics of death of *E. coli* cells (in % of CFU) at various concentrations of CFCs. (**A**) CFC-1, (**B**) CFC-2.

**Figure 7 materials-16-00957-f007:**
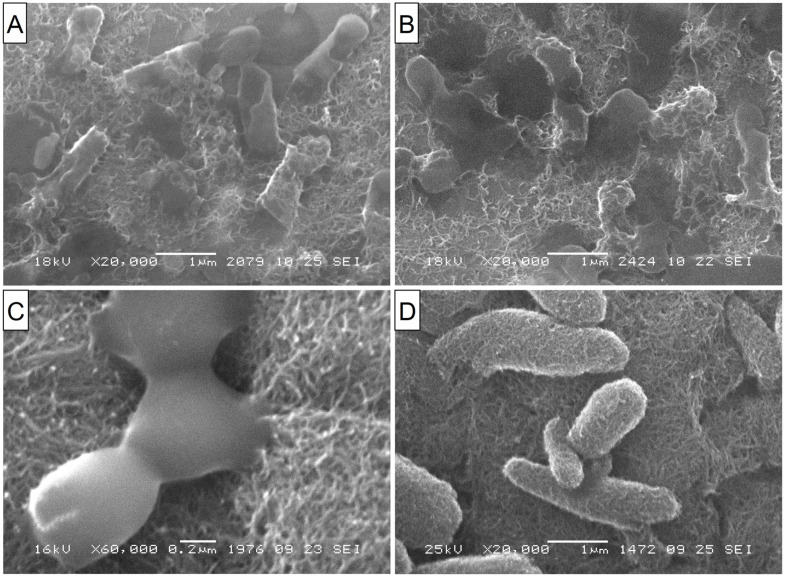
(**A**) SEM images of *E. coli* adsorbed on carboxylated MWCNT-1ox (0.6) after 24-h exposure. Significant structural destruction of the cell walls and membranes is seen. (**B**) Debris of completely destroyed *E. coli* cells on the surface of the MWCNT-2ox (0.5) aggregate. (**C**,**D**) SEM images of S. aureus on the surface of MWCNT-2ox (0.5) aggregates. One can see the enveloping of the cells with nanotubes, followed by their further damage.

**Figure 8 materials-16-00957-f008:**
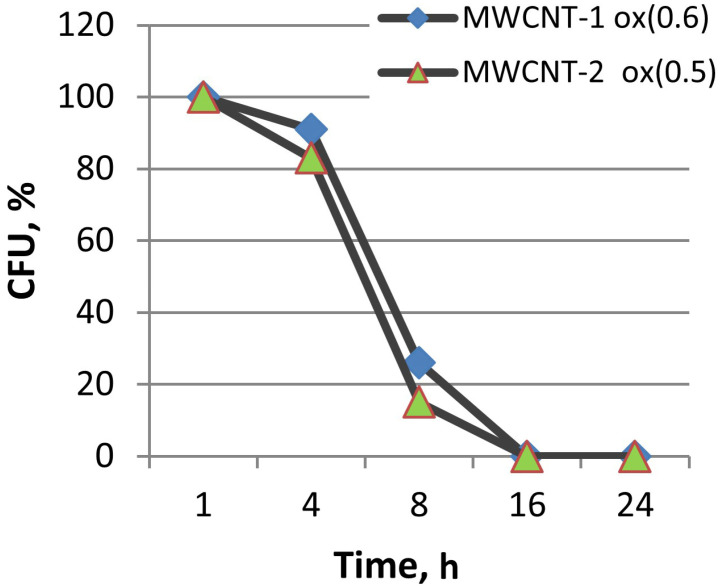
A decrease in the number of *E. coli* colony-forming units over time in the presence of stable suspensions of carboxylated MWCNTs.

**Figure 9 materials-16-00957-f009:**
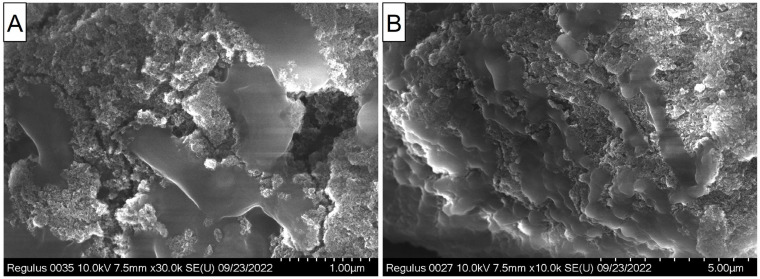
SEM images of ND aggregates with the adsorbed and morphologically distorted *E. coli* cells.

**Figure 10 materials-16-00957-f010:**
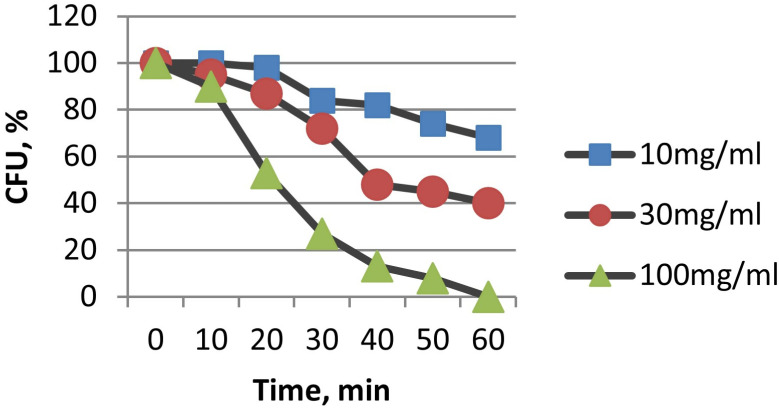
Influence on the kinetics of death of *E. coli* cells (in % of CFU) at different concentrations of NDs.

**Figure 11 materials-16-00957-f011:**
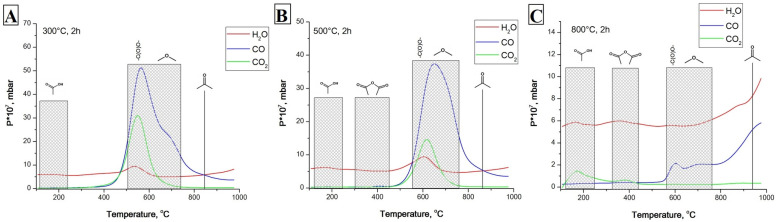
Temperature-programmed desorption spectra (mass-spectrometric analysis of the evolution of H_2_O, CO, and CO_2_ during heating of CFC-2 in vacuum) of CFC-2 pretreated in H_2_ flow at 300 °C (**A**), 500 °C (**B**), and 800 °C (**C**). One can observe that hydrogen pretreatment results in the elimination of oxygenated surface groups from CFC-2. NDs and MWCNT-1 ox (0.5) demonstrated similar behavior concerning the elimination of oxygen-containing groups.

**Figure 12 materials-16-00957-f012:**
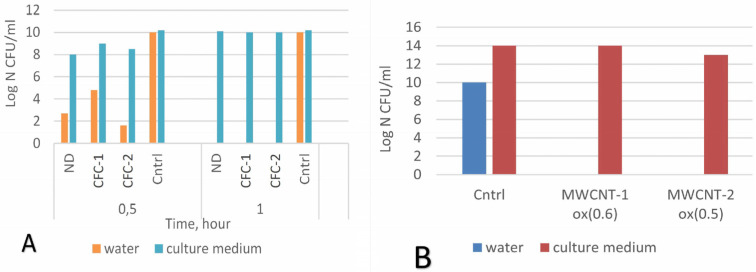
Influence of the addition of peptone and glucose to the liquid medium on the antimicrobial effects of CNMs (after 24 h incubation). (**A**) NDs and CFCs. (**B**) MWCNT 1(2)ox.

**Figure 13 materials-16-00957-f013:**
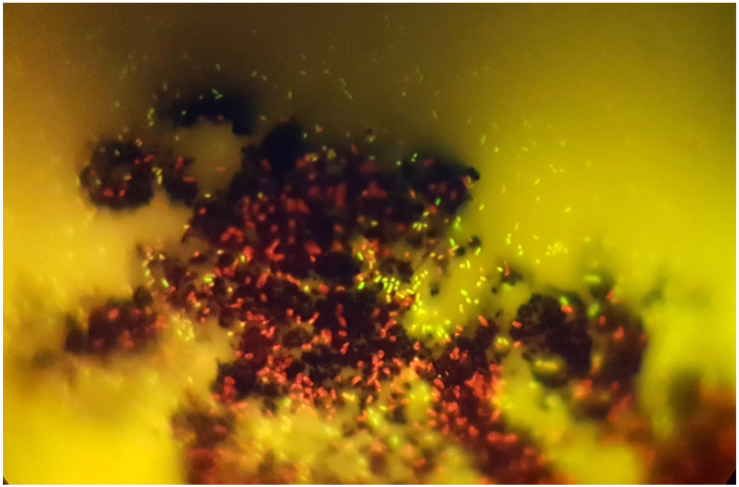
Viability measurement of *E. coli* after 24-h exposure to MWCNT-1ox. Fluorescence-based assay shows the antimicrobial activity of MWCNTs: dead cells stained with PI (red) are adsorbed on MWCNTs, while live cells (yellow) are located between nanotube aggregates.

**Table 1 materials-16-00957-t001:** Structural/textural characteristics and elemental composition of nanocarbons.

	Nanocarbons *
	Nano-Diamonds(ND)	Onion-like Carbon(OLC)	Carbon Nanofibers(CFC-1),	Carbon Nanofibers (CFC-2)	MWCNTs	MWCNT-ox
**Structure**	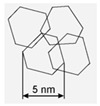	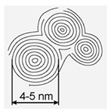	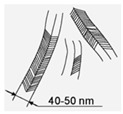	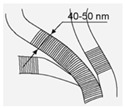	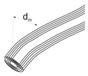	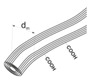
**S_BET_**(m^2^/g)	350–370	500	90–110	140–160	**MWCNT-1 360**(d = 7.8 nm, L = 40 µm)**MWCNT-2 260**(d = 10 nm, L = 30 µm)**MWCNT-3 120**(d = 18 nm, L = 20 µm)	**MWCNT-1ox (0.6)** **340–350** **MWCNT-2ox (0.5)** **250–280**
V pore, m^3^/g	0.79	1.09	0.29	0.43	1.5–1.65	1.5–1.8
Number of –COOH/nm^21^	3.1	<0.1	11.9	7.8	0.2–0.8	2.1–2.4
Carbon content, %	>99.9	>99.9	>99.5	>99.5	>99.5	>99.5

* Carboxyl-functionalized CFC-1, MWCNT-2ox, and NDs heated in H_2_ at 300, 500, and 800 °C (for variation in surface oxygenated groups) were also used for comparison purposes.

**Table 2 materials-16-00957-t002:** A comparison of antibacterial properties of oxygen-containing and deoxygenated CNMs.

CNM	Content of COOH/1 nm^2^	CFU % after 24 h Exposure of *E. coli* and CNMs	CFU % after 24 h Exposure of *E. coli* and CNMs Treated at 800 °C in H_2_
CFC-2	7.78	0	0
MWCNT-2ox (0.5)	2.13	0	0
NDs	3.13	0	0

## Data Availability

The evaluated data presented in this study are available in the tables of this paper. The raw measured data of this study are available on request from the corresponding author.

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
