# Peer review of "Antibacterial Effect of Carbon Nanomaterials: Nanotubes, Carbon Nanofibers, Nanodiamonds, and Onion-like Carbon"

_materials, 2023, doi:10.3390/ma16030957_

Round 1

Reviewer 1 Report

The paper deals with an interesting topic that is considered of high concern for the academic and non academic community.

Nevertheless, some issues need to be better addressed by the Authors considering the huge amount of the scientific literature in the field.

Specifically, they investigate the antimicrobial perfomance of the proposed nanomaterials performing the experiments in water (to avoid aggregation/modification of the CNMs), but this medium is not appropriate for the microorganisms proliferation and growth...in fact, adding peptone and glucose they obtain significant differences in terms of antimicrobial efficacy of the CNMs. They should perform their tests in experimental conditions able to preserve the microorganisms environment in order to correctly evaluate the real CNMs effectiveness.

Also, the nanotoxicity issues should be taken under proper consideration. please commet on this.

Which are the potential application field of the proposed nanomaterials? This aspect is essential to evaluate the significance of the proposed approach.

The references must be revised to include relevant papers on the antimicrobial nanomaterials and antibiotic/antimicrobial resistance.

Minor aspects:

-the English should be revised

-Table 1 formatting should be checked

On these basis I propose to reconsider the paper after addressing the reported comments/revisions.

Author Response

Response to First Referee

Dear Sir,

We express our sincere gratitude to the referee for his interest in the work and comments that help improve the article.

Referee 1

The paper deals with an interesting topic that is considered of high concern for the academic and non academic community.

Nevertheless, some issues need to be better addressed by the Authors considering the huge amount of the scientific literature in the field.

  1. Specifically, they investigate the antimicrobial perfomance of the proposed nanomaterials performing the experiments in water (to avoid aggregation/modification of the CNMs), but this medium is not appropriate for the microorganisms proliferation and growth...in fact, adding peptone and glucose they obtain significant differences in terms of antimicrobial efficacy of the CNMs. They should perform their tests in experimental conditions able to preserve the microorganisms environment in order to correctly evaluate the real CNMs effectiveness.

Response :

In most of the previously published works, research on the study of the antimicrobial properties of materials is carried out exclusively in water. Water is an ideal medium for the experiment, since both reproduction and death of bacteria are excluded in it in the time range chosen by the experiment, and at the same time, there is no effect of impurities on the structure and properties of nanomaterials. We also conducted studies both in water and in nutrient broth, which showed the dependence of the antimicrobial action on the composition of the solutions, which is explained by the possibility of blocking the centers that firmly bind microorganisms and cause the destruction of their walls.

These observations indicate the need to develop antibacterial systems using several components. In particular, combinations of CNMs with coated silver particles or antibiotics may be of interest. At the same time, the identified CNMs with a high bactericidal effect can also be used as components of air-cleaning devices, membranes, antibacterial fabric materials and polymer coatings, where the influence of liquid media is significantly limited.

We have made appropriate corrections to the conclusion.

  1. Also, the nanotoxicity issues should be taken under proper consideration. please commet on this.

Nanotoxicology focuses on determining the adverse effects of nanomaterials on human health and the environment. The use of carbon nanomaterials, due to the nature of their properties, also requires careful study of their toxicity. Many works are devoted to these issues, we limited ourselves to adding a four reviews (see also the refs [13, 23 ]).

Asati S., Sahu A. and h Jain A., Nanotoxicity: The Dark Side of Nanoformulations. In Current Nanotoxicity and Prevention, Vol. 1, Issue 1, 2021, 6 – 25,  DOI: 10.2174/2665980801999201230095324

Kobayashi, N., Izumi, H., & Morimoto, Y. (2017). Review of toxicity studies of carbon nanotubes. Journal of Occupational Health, 59(5), 394–407. doi:10.1539/joh.17-0089-ra

Yuan, X., Zhang, X., Sun, L., Wei, Y., & Wei, X. Cellular Toxicity and Immunological Effects of Carbon-based Nanomaterials. Particle and Fibre Toxicology, 2019, 16:18. doi:10.1186/s12989-019-0299-z 

Madannejad, R., Shoaie, N., Jahanpeyma, F., Darvishi, M. H., Azimzadeh, M., & Javadi, H. (2019). Toxicity of carbon-based nanomaterials: Reviewing recent reports in medical and biological systems. Chemico-Biological Interactions. doi:10.1016/j.cbi.2019.04.036.

In our case, we did not consider the issues of nanotoxicity of the used NCMs, since the aspects of their application that we are developing do not imply their introduction into the human body. First of all, we assume the development of devices that provide air disinfection, filter materials and antimicrobial coatings. In this case, it is supposed to provide a sufficiently strong bond of the NCM with the structural materials of the base. In this case, even with the destruction of such materials, the toxicity of the products of their destruction will be close to the toxicity of widespread carbon materials (carbon blacks, coals), with which mankind has been in contact throughout its history.

Following your comment, we have included some small remarks about the nanotoxicity of nanomaterials in the conclusion.

  1. Which are the potential application field of the proposed nanomaterials? This aspect is essential to evaluate the significance of the proposed approach.

Response: Please see the responses 1 and 2.

We have made appropriate corrections to the conclusion.

  1. The references must be revised to include relevant papers on the antimicrobial nanomaterials and antibiotic/antimicrobial resistance.

Paper contains numerical references concerning of antibacterial activity of nanocarbons [8-25,40-54].

There are several reviews among them. Among them there are several reviews. It's hard to know exactly which additional refs. you offered.

Minor aspects:

  1. -the English should be revised

Response: The English has been corrected with the help of a professional translator.

  1. Table 1 formatting should be checked

Response: We removed editorial errors from Table 1.

Reviewer 2 Report

Dear,

In this study, within a single methodology, authors try to determine the presence of  antimicrobial properties of a wide systemic set of CNMs (Multiwalled carbon nanotubes (MWNTs), catalytic carbon nanofibers with different orientations of graphene blocks (coaxial-conical and stacked, CFC), ionic carbon (OLC), and ultrafine explosive nanodiamonds (NDs) were used as a system set of CNMs) . They compare the effects caused by different types of CNMs on typical common opportunistic pathogens, namely Escherichia coli (hereinafter E. coli) and Staphylococcus aureus (hereinafter S. aureus). Their study showed that the antimicrobial effect of CNMs strongly depends on their structure, particle size, aggregation degree, concentration, and surface functionalization. Thus, onion-like carbons did not show antimicrobial properties, possibly due to the low concentration of surface defects and hydrophobicity, which led to their low ability to adsorb microorganisms. At the same time, nanodiamonds and catalytic fibrous carbon, and to a lesser extent carboxylated MWCNTs, have an antimicrobial effect.

 Authors, also showed, the data obtained indicate the promise of using a number of CNMs for the development of membranes and polymer coatings containing their inclusions for the creation of materials with antimicrobial properties.

It is evident that the great experience and work of the authors stand behind this research, but I still have some comments and suggestions:

Due to the large number of abbreviations, the manuscript is very difficult to read, that is, to follow how the tested carbon nanomaterials (CNMs) react in certain conditions.

1.       Line 20: NA is abbreviation of what?

2.       Line 248-249: This is confirmed by the data of fluorescence microscopy with staining of cells with CYTO9 dye (Figure 5, C and D). Please, explain CYTO9 dye?

3.       You use a lot of abbreviations that seem confusing in your manuscript. In addition, for many abbreviations, you do not display the full names (CFU, CYTO9, CCMs), and you often make mistakes in specifying the abbreviations (NCMs-Line 340).

4.       Line 417-418: In conclusion there is sentence: The authors of the NS review suggest that the toxicity dose of CNT IC50 is about 4–5 mg ml-1 h-1.

It is not clear which authors you are talking about in this sentence. If this is information/data from the literature, you must provide the reference.

NS review? What is it?

I suggest you to reduce the number of abbreviations and use full names of substances.

Finally, I would ask you to explain what is the novelty of this (yours) research?

Sincerely

Author Response

Dear Sir,

We express our sincere gratitude to you for your interest in the article, the time spent and comments that helped improve the article. We have taken into account almost all your comments. First of all, we corrected all errors in the presentation of abbreviations for the names of nanomaterials. All fixes are highlighted in yellow.

Comments and answers.

  1. Line 20: NA is abbreviation of what?

Response: It was an unfortunate typo, changed to NDs (Nanodiamonds)

  1. Line 248-249: This is confirmed by the data of fluorescence microscopy with staining of cells with CYTO9 dye (Figure 5, C and D). Please, explain CYTO9 dye?

Response: It was an unfortunate typo, changed in the text to SYTO 9.  SYTO™ 9 Green Fluorescent Nucleic Acid Stain is an excellent green-fluorescent nuclear and chromosome counterstain that is permeant to both prokaryotic and eukaryotic cell membranes.

  1. You use a lot of abbreviations that seem confusing in your manuscript. In addition, for many abbreviations, you do not display the full names (CFU, CYTO9, CCMs), and you often make mistakes in specifying the abbreviations (NCMs-Line 340).

Response: I am sorry for typos. All necessary amendments have been done:

Catalytic filamentous carbon (CNF)

Carbon nanomaterials (CNMs)

Colony-forming unit (CFU).   Thus,  0.5 McFarland turbidity standard provides an optical density comparable to the density of a bacterial suspension with a 1.5 x 10^8 colony forming units (CFU/ml).

  1. Line 417-418: In conclusion there is sentence: The authors of the NS review suggest that the toxicity dose of CNT IC50 is about 4–5 mg ml-1 h-1.

It is not clear which authors you are talking about in this sentence. If this is information/data from the literature, you must provide the reference.

NS review? What is it? I suggest you to reduce the number of abbreviations and use full names of substances.

Response:  Corrected.

The authors of the review [13], after analyzing a large number of articles, concluded that the toxic dose of CNTs  IC50 is about 4–5 mg ml-1 h-1. Based on our data, we also suggest that the toxicity range of MWCNT-ox is comparable to this value. 

5.Finally, I would ask you to explain what is the novelty of this (yours) research?

Response:  

As for the novelty of this work...

A large number of works have appeared on the antimicrobial effects of CNMs. But in the vast majority of studies, the authors study only one type of CNM. All these studies were carried out using completely different methods, and in some cases the conclusions contradict each other. In our study, we took a systemic set of CNMs and compared the antimicrobial effects using a single, and in our opinion, the simplest and most reliable method (excluding any instrument errors) - the method of seeding on nutrient agar in Petri dishes, in which the ability of microorganisms to form colonies is visually visible. The experiment on antimicrobial effects was confirmed by microscopic images.

In addition, in most previously published works, studies are carried out exclusively in water. Water is an ideal environment for the experiment, since in it, in the time range chosen by the experiment, both reproduction and death of bacteria are excluded, and at the same time, there is no effect of impurities on the structure and properties of nanomaterials. We conducted studies both in water and in nutrient broth, showing the dependence of antimicrobial effects on the composition of the solutions.

It has been established that the presence of a nutrient medium largely suppresses the antibacterial properties of CNMs, which indicates the need to develop antibacterial systems using several components. At the same time, the identified CNMs with a high bactericidal effect can also be used as components of air-cleaning devices, membranes, antibacterial fabric materials and polymer coatings, where the influence of liquid media is significantly limited.

We have included appropriate corrections to the conclusion.

All corrections marked in yellow.

Round 2

Reviewer 1 Report

The Authors properly answered/commented all the points raised by the Reviewer.